# Thyroid Cancer Detection using Smartphone-captured Cytopathology Images and Foundation Model

**Vladimir Cuc**[1]                                                                     VCUC@MOCS.FLSOUTHERN.EDU
**Aiden Coffey**[1]                                                                    ACOFFEY@MOCS.FLSOUTHERN.EDU
**Hoan Thanh Ngo**[1]                                                                 HNGO@FLSOUTHERN.EDU
**Walter Lee**[2]                                                                      WALTER.LEE@DUKE.EDU
[1] *Florida Southern College, Lakeland, FL, USA*
[2] *Duke University Medical Center, Durham, NC, USA*

**Editors:** Under Review for MIDL 2025

## Abstract

Recent foundation models for pathology—such as UNI (Chen et al., 2024), CONCH (Lu et al., 2024), RudolfV (Dippel et al., 2024), Prov-GigaPath (Xu et al., 2024), Atlas (Alber et al., 2025), and Virchow2 (Zimmermann et al., 2024)—have demonstrated impressive performance, but are typically trained on images captured by costly whole-slide imaging scanners. In contrast, many hospitals in developing countries still rely on optical microscopes and low-cost cameras or smartphones for image acquisition. To bridge the gap, we demonstrate that pathology foundation model pre-trained on whole-slide images (WSIs) can be fine-tuned on smartphone-captured cytopathology images for applications in low-resource settings. We used over 3,000 smartphone-captured cytology images from Tanzania and Vietnam for Virchow2 foundation model fine-tuning and testing. Our approach not only resulted in high classification performance (98.17% AUC, 92.93% accuracy, 94.13% F1-score) but also enhanced interpretability through principle component visualization of the embedding space, thereby fostering clinical trust in resource-constrained settings.

**Keywords:** thyroid cancer, computer aided-diagnosis, explainable AI, smartphone, Vision Transformer, low-cost digital pathology, foundation model

## 1. Introduction

Globally, the incidence of thyroid cancer is on the rise, yet many regions still struggle with high diagnostic costs. In low- and middle-income countries, the expensive nature of whole-slide imaging scanners limits slide digitization and application of computer-aided diagnosis. We previously proposed a low-cost workflow that utilizes a smartphone attached to an optical microscope and CNN for digitizing and analyzing cytopathology slides (Assaad et al., 2023). In this work, we fine-tuned Virchow2, a foundation model for pathology, on smartphone-captured cytopathology images for thyroid cancer detection. Beyond delivering robust classification performance, this approach generates visual explanations through heatmaps, thereby supporting clinical integration and enhancing transparency.

## 2. Methods

### 2.1. Datasets and Preprocessing

We collected a total of 3,246 smartphone-captured cytopathology images from two hospitals: 1,229 Pap-stained images from 100 patients at Kilimanjaro Christian Medical Centre

(Tanzania), and 2,017 alcohol-fixed images from 132 patients at the National ENT Hospital (Vietnam) with IRB approvals. All images were captured using Redmi Note 10S smartphones mounted on microscopes. Ground truth labels were determined based on surgical outcomes. Preprocessing steps included stain normalization, cropping, and resizing to 224×224 pixels. The dataset was then stratified and split per patient, into a train set (70%), a validation set (20%), and a test set (10%).

## 2.2. Model and Training

Our core model, Virchow2, is a foundation model for pathology (Zimmermann et al., 2024). Pre-trained on 3.1 million WSIs, we custom-trained it on our smartphone-captured datasets. We used the Adam optimizer with cosine decay, data augmentation (color jitter, rotation, zoom), and early stopping based on validation AUC. The full pipeline is shown in Fig. 1.

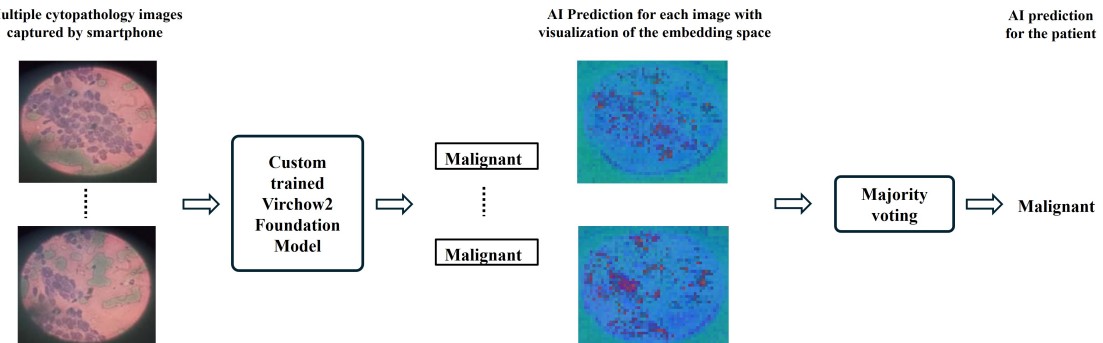

Figure 1: We custom-trained the Virchow2 pathology foundation model on smartphone-captured cytopathology images for application in low-resource settings.

## 2.3. Baseline Models

For comparative analysis, we also trained EfficientNetB0 CNN (Tan and Le, 2019), Google ViT (Vision Transformer) (Steiner et al., 2022), and Meta's DINOv2 self-supervised vision transformer (Maxime Oquab, 2023) models using identical splits and evaluation metrics.

## 2.4. Heatmap Analysis Techniques

To enhance model interpretability, we compared two heatmap generation methods: (1) Visualizing the embedding space using Principle Component Analysis (PCA) (Maxime Oquab, 2023) and (2) Occlusion-based method using Captum library (Kokhlikyan et al., 2020)

## 3. Results and Discussion

### 3.1. Testing Performances

Using AUC as our primary metric, Virchow2-based model achieved the highest AUC (98.17%) as shown in Table 1. We believe that, in the next phase of our study, it will generalize better than the other models since it was pre-trained on 3.1M WSIs.

| Model | AUC | Accuracy | F1-Score | Precision | Recall |
|---|---|---|---|---|---|
| Google_ViT-based | 91.59% | 91.63% | 92.34% | 91.64% | 93.17% |
| Efficientnet-B0-based | 98.15% | 93.23% | 94.07% | 90.83% | **97.54%** |
| DINOv2-based | 97.76% | **93.57%** | 93.37% | **93.47%** | 96.47% |
| Virchow2-based | **98.17%** | 92.93% | **94.13%** | 91.11% | 96.65% |

Table 1: Test performance comparison of models on thyroid cancer classification.

### 3.2. Explainability

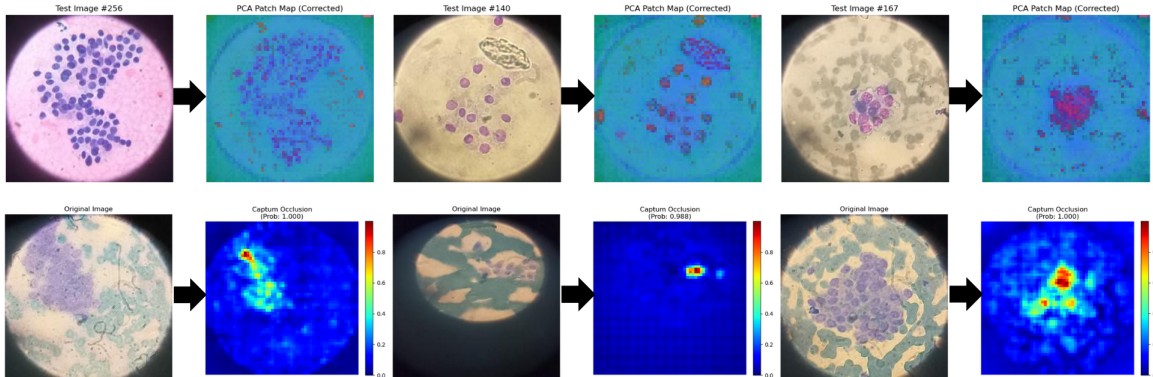

Figure 2: Heatmaps generated by visualization of the embedding space using PCA (top row) and occlusion-based method using Captum library (bottom row).

We tested two heatmap generation methods to provide insights into the custom-trained Virchow2 model's decision process. The top row of Fig. 2 shows heatmaps generated by PCA visualization of the patch features output by the model. The heatmaps show that the model can separate the image patches into meaningful clusters and can differentiate parts of the cell. The bottom row of Fig. 2 displays heatmaps produced by an occlusion-based method using Captum library. The heatmaps highlight regions where occlusion causes a significant drop in classification confidence, revealing key areas influencing the model's decision.

### 4. Conclusion

Our experiments demonstrate that Virchow2 foundation model fine-tuned on smartphone-captured cytopathology images not only excels in accuracy but also provides robust visual interpretability. PCA visualization of the embedding space showed finer-grained features down to the cellular level, bolstering confidence in the model's ability to localize diagnostically relevant features and increasing its potential for clinical adoption.

**Acknowledgments** The authors graciously acknowledge funding from the National Institutes of Health (1R21CA268428-01 and 4R33CA268428-03). We also thank Dr. TrungNguyen, Dr. Luan Nguyen, Dr. Alex Mremi, and Daniel Mbwambo for data collection.

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
