# OpenReview forum: "Thyroid Cancer Detection using Smartphone-captured Cytopathology Images and Foundation Model"
_MIDL.io/2025/Short_Papers — MIDL 2025 - Short Papers_

### Official Review · Reviewer_Y2wG · 2025-04-28

**Rating:** 3
**Confidence:** 4

**Summary:**

This work explored fine-tuning a pathology foundation model trained on WSI (Virchow2) to a dataset of smartphone-captured cytology images from Tanzania and Vietnam for thyroid cancer detection. The method was compared to using natural image pretrained models to start. Interpretation of the model results was visualized using two interpretation methods.

**Strengths:**

1. This work addresses an important problem - application of well pretrained models to low-resource settings, where only smartphone captured pathology images are available.

2. The work includes comparisons to using other common foundation models pretrained on natural images.

3. The experiment utilized 2 datasets from 2 low-resource populations, and showed the approach and overall high performance.

**Weaknesses:**

1. Given the single test data split (many samples but from ~20 subjects) and the performance of different methods reported in Table 1, I would say that there is not a significant difference between the proposed pathology-based foundation model and 2 of the 3 natural image models (EfficientNet B0 and Dinov2) - there is only 0.02 difference in AUC, 0.05 difference in F-1 score, and recall is higher in EfficientNet, while Dinov2 showed highest accuracy and precision. Therefore it is difficult to say that the proposed method performs the best.

2. The proposed method was compared to multiple natural image foundation models, but why not any of the other pathology foundation models, many of which are mentioned in the introduction? This would have provided an interesting study on generalizability of different pathology foundation models.

3. The qualitative interpretation results are shown only for the proposed method - as the interpretation methods can be applied post hoc to any model, it would have been helpful to see how the explanations compared between the different tested models. If the performance was similar but the explanations looked more meaningful for the proposed approach, then this would provide more weight for the statement that the proposed method enhances interpretability.

Minor: "principle" should be "principal" component analysis

---

### Decision · Program_Chairs · 2025-05-01

Accept